# Human-Guided Design to Explain Deep Learning-based Pneumothorax Classifier

**Han Yuan**[1]                                                    YUAN.HAN@U.DUKE.NUS.EDU
**Peng-Tao Jiang**[2]                                                    PT.JIANG@ZJU.EDU.CN
**Gangming Zhao**[3]                                              GMZHAO@CONNECT.HKU.HK

[1] *Duke-NUS Medical School, National University of Singapore*

[2] *College of Computer Science and Technology, Zhejiang University*

[3] *Faculty of Engineering, The University of Hong Kong*

**Editors:** Accepted for MIDL 2023

## Abstract

Pneumothorax (PTX) is an acute thoracic disease that can be diagnosed with chest radiographs. While deep learning (DL) models have proven effective in identifying PTX on radiographs, they have difficulties in gaining the trust of radiologists if the decision-making logic is unclear. Therefore, various methods have been proposed to explain the PTX diagnostic decision made by DL models. However, several studies indicate that the quality of DL model explanation is suboptimal. This paper introduces a human-guided approach to enhance the existing explanation method. Based on the IoU and Dice between the explanation of model-focusing regions and the ground truth lesion areas, we achieved an increase of 60.6% and 56.5% in Saliency Map, 69.0% and 66.7% in Grad-CAM, and 137.5% and 123.9% in Integrated Gradients.

**Keywords:** Pneumothorax Diagnosis, Convolutional Neural Network, Explainable Artificial Intelligence, Human-in-the-loop

## 1. Introduction

Pneumothorax (PTX) is an acute thoracic disease caused by the abnormal air collection in the pleural space (the space between the lungs and chest wall) (Imran and Eastman, 2017). Prompt treatment of PTX prevents it from progressing into a life-threatening emergency (Thian et al., 2021). In clinical practice, radiologists use chest radiographs (chest X-rays) to facilitate PTX diagnosis, which requires a great deal of human effort. According to several recent studies, such a process can be automated using deep learning (DL) models, especially the convolutional neural network (CNN) (Thian et al., 2021). These CNN-based classifiers have displayed high-fidelity PTX classification capability while they use a large number of interconnected neurons whose relationships are highly complex and difficult to comprehend. Thus, these diagnostic decisions have difficulties in gaining the trust of radiologists (Rudin, 2019; Xie et al., 2022).

To solve this problem, researchers incorporated various explanation methods for chest radiograph analysis to highlight the areas on chest radiographs contributed most to the disease diagnosis (Van der Velden et al., 2022). However, a recent benchmarking study pointed out that a sophisticated CNN achieved an AUROC of 0.993 in the PTX classification while its focus area (model explanation) generated by Integrated Gradients (Sundararajan et al., 2017) only overlapped with 7% of the ground truth lesion area (Saporta et al., 2022).

Therefore, there is a demand to develop improved model explanation methods (Saporta et al., 2022).

The inclusion of prior expert knowledge in the model explanation process is one promising direction for advancement. For example, PTX frequently occurs in the pleural space between the lungs and chest wall (Imran and Eastman, 2017). An intuitive hypothesis is that the location information of the pleural space could contribute to the explanation of PTX classifiers. In this study, we propose a heuristic method that extracts the PTX high occurrence area (the pleural space) from a few lesion-delineated PTX cases and uses that information to guide model explanations.

## 2. Methods and Experiments

In image classification, the explanation paradigm calculates each pixel's importance towards the model prediction and outlines the sub-regions (focus area) consisting of the most important pixels (Zhou et al., 2016; Van der Velden et al., 2022). The model is considered trustworthy if its focus area is precisely matched with the area on which human experts make decisions. In this study, we implemented three popular techniques of Saliency Map (Simonyan et al., 2013), Grad-CAM (Selvaraju et al., 2017), and Integrated Gradients (Sundararajan et al., 2017) to generate the model explanation (focus area) and illustrate the efficacy of our proposed method as a plug-and-play module.

Based on the prior clinical knowledge (Imran and Eastman, 2017), we propose incorporating the disease occurrence area into the existing explanation methods. Specifically, using one canonical PTX delineation selected by human experts as the starting point, the PTX template is generated by horizontal flipping, overlap, and dilation. Horizontal flipping and overlap aim to spotlight both left and right pleural spaces while the dilation step enlarges the template to cover the broader pleural space. The template is then laid over the original model explanation to focus on the pleural space. Figure 1 compares the baseline and our enhanced explanations. Besides, affine transformation (Liu et al., 2019) is implemented to eliminate the deformation such as the improper distance, angle, and displacement in the original radiographs, and further upgrade the effectiveness of our method.

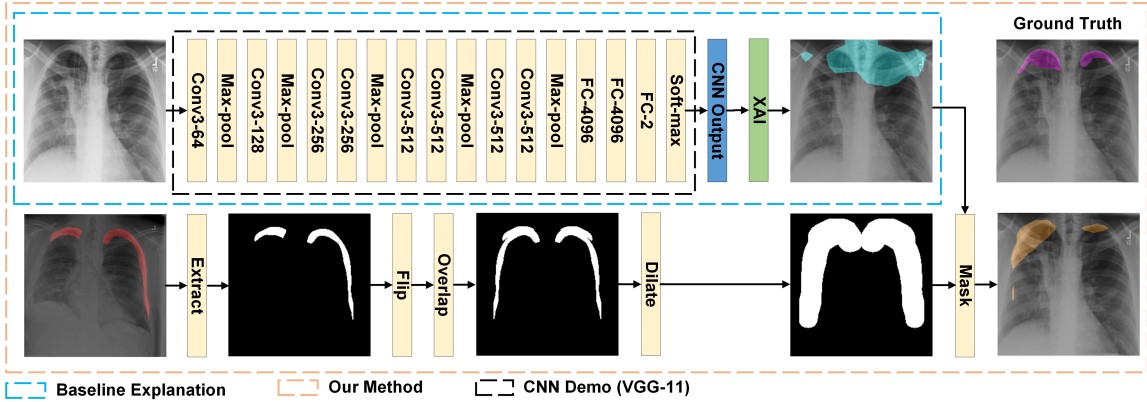

Figure 1: Comparison of the baseline explanation and our proposed method

We developed the PTX classifier with a light-weighted backbone of VGG-11 (Simonyan and Zisserman, 2015) and modified its output layer into 2 to comply with the binary diagnosis. Stochastic Gradient Descent (Rumelhart et al., 1986) was used as the optimizer, with an initial learning rate of 1e-3 and a momentum of 0.9. The learning rate was scheduled with a reduction factor of 0.5 if no improvement was observed for 3 epochs. Model training was conducted in batches of 16 images and the loss was measured by weighted cross-entropy. The training epoch was set as 50 along with early stopping evaluated on the validation set. After the completion of model training, the classification performance was evaluated by AUROC, AUPRC, accuracy, sensitivity, and specificity on the unseen test dataset.

After the binary PTX classification training, three CNN explanation methods were utilized. With our method and affine transformation as plug-and-play modules on the existing methods, we had a total of 12 explanation methods. The direct production of these methods was the pixel importance, and the focus area is extracted as the final explanation. The model explanation of the focus area was evaluated through the IoU and Dice score coefficient (Dice) on the ground truth lesion area of PTX-positive samples in the test dataset (Liu et al., 2019). The 95% CIs was computed by bootstrapping (Efron, 1987).

## 3. Results

Based on the SIIM-ACR Pneumothorax Segmentation Challenge dataset[1], we analyzed 12,047 chest radiographs (containing 2,668 PTX-positive cases) in terms of the PTX diagnoses and used 60/20/20 random splitting to generate the training, validation, and test set. All radiographs contained binary diagnostic labels for the PTX classifier training and testing. Ten PTX samples in the validation set contained pixel-level lesion annotations for the focus area generation and all PTX samples in the test set included pixel-level lesion annotations for the focus area evaluation.

In PTX binary classification, the classifier trained on affine-transformed datasets achieved marginally better results of an AUROC of 86.4% (±1.7), an AUPRC of 68.0% (±4.0), an accuracy of 80.0% (±1.7), a sensitivity of 76.8% (±3.4), and a specificity of 80.9% (±2.0). With well-trained classifiers and original explanation methods, Figure 2 summarizes an ablation study to verify the explanation improvements. Adding either affine transformation or our method improved all three explanation methods, while the use of both resulted in more prominent improvements in terms of IoU and Dice: 60.6% and 56.5% for Saliency Map, 69.0% and 66.7% for Grad-CAM, and 137.5% and 123.9% for Integrated Gradients.

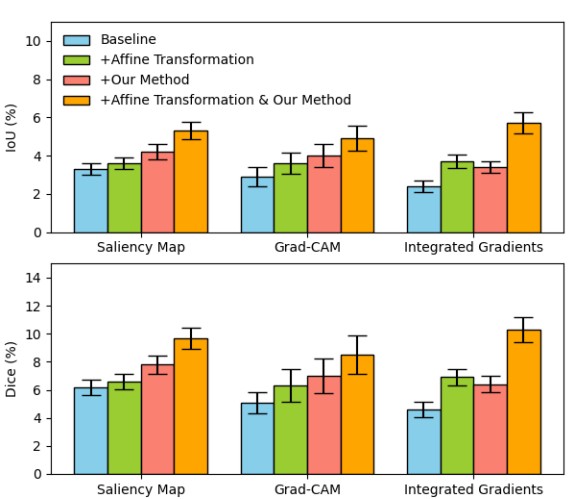

Figure 2: Explanation result comparison

---

1. https://www.kaggle.com/c/siim-acr-pneumothorax-segmentation
   https://www.kaggle.com/datasets/vbookshelf/pneumothorax-chest-xray-images-and-masks

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
