# OpenReview forum: "Human-Guided Design to Explain Deep Learning-based Pneumothorax Classifier"
_MIDL.io/2023/Short_Paper_Track — MIDL 2023 Short paper track Poster_

### Official Review · Reviewer_Gkh9 · 2023-04-21
**Reivew**

**Rating:** 7
**Confidence:** 4

**Review:**

This paper introduces a human-guided approach to enhance the existing explanation method. The paper is generally well written with the methods being easy to understand and follow.
However, there are some minor issues needed to be addressed in the revision:
1. More details of prior clinical knowledge should be provided.
2. It will be better to provide some visual results.

---

### Official Review · Reviewer_orak · 2023-04-22
**A generic method to include human priors on top of XAI methods, that may however backfire**

**Rating:** 5
**Confidence:** 3

**Review:**

The paper proposes a generic method to improve overlap of explanability maps with plausible anatomical regions derived from human expertise.

Pros:
- The method is generic and can apply to different XAI techniques
- Large and open data is used to evaluate the method
- Simple way to include human expertise as location priors

Cons:
- The method works by incoporating a strong prior that pleural space is the location for PTX, which should be the case for the majority of the dataset.
Given this, it is almost by definition that the metrics would improve since ground truth annotations will focus on pleural space. But this comes with a disadvantage - what if other subtle signs of the disease are present (along the lines of ground glass opacity, or other radiological signs - not saying that GGO is a feature of PTX, this is an example) and picked up by the classifier? They would then be removed by the very strong prior, and this would arguably distort the explanation by removing parts the image the network is actually focusing on. This would also prevent debugging egregious mistakes like for instance using pixels with burn-in anntations for classification - the fact that the classifier is exploiting this type of regularity would then be hidden because the mask only shows explanations in the a-priori plausible locations.
- Numeric results are unclear - why do integrated gradients lead to > 100% results? Figure 2 - what is the y axis scale showing ? This should not be improvement since the baseline is shown?
- No code available